# A Cost-Effective Model for Predicting Recurrent Gastric Cancer Using Clinical Features

**DOI:** 10.3390/diagnostics14080842

**Published:** 2024-04-18

**Authors:** Chun-Chia Chen, Wen-Chien Ting, Hsi-Chieh Lee, Chi-Chang Chang, Tsung-Chieh Lin, Shun-Fa Yang

**Affiliations:** 1Institute of Medicine, Chung Shan Medical University, Taichung 40201, Taiwan; chenjica@gmail.com (C.-C.C.); ysf@csmu.edu.tw (S.-F.Y.); 2Division of Plastic Surgery, Department of Surgery, Chi Mei Medical Center, Tainan 704, Taiwan; 3Division of Colorectal Surgery, Department of Surgery, Chung Shan Medical University Hospital, Taichung 40201, Taiwan; cshy1294@csh.org.tw; 4School of Medicine, Chung Shan Medical University, Taichung 40201, Taiwan; 5Department of Computer Science and Information Engineering, National Quemoy University, Kinmen County 892, Taiwan; s110740704@student.nqu.edu.tw; 6School of Medical Informatics, Chung Shan Medical University & IT Office, Chung Shan Medical University Hospital, Taichung 40201, Taiwan; 7Department of Information Management, Ming Chuan University, Taoyuan City 33300, Taiwan

**Keywords:** recurrent gastric cancer, random forest, SMOTE, SHAP, cost-sensitive learning

## Abstract

This study used artificial intelligence techniques to identify clinical cancer biomarkers for recurrent gastric cancer survivors. From a hospital-based cancer registry database in Taiwan, the datasets of the incidence of recurrence and clinical risk features were included in 2476 gastric cancer survivors. We benchmarked Random Forest using MLP, C4.5, AdaBoost, and Bagging algorithms on metrics and leveraged the synthetic minority oversampling technique (SMOTE) for imbalanced dataset issues, cost-sensitive learning for risk assessment, and SHapley Additive exPlanations (SHAPs) for feature importance analysis in this study. Our proposed Random Forest outperformed the other models with an accuracy of 87.9%, a recall rate of 90.5%, an accuracy rate of 86%, and an F1 of 88.2% on the recurrent category by a 10-fold cross-validation in a balanced dataset. We identified clinical features of recurrent gastric cancer, which are the top five features, stage, number of regional lymph node involvement, *Helicobacter pylori*, BMI (body mass index), and gender; these features significantly affect the prediction model’s output and are worth paying attention to in the following causal effect analysis. Using an artificial intelligence model, the risk factors for recurrent gastric cancer could be identified and cost-effectively ranked according to their feature importance. In addition, they should be crucial clinical features to provide physicians with the knowledge to screen high-risk patients in gastric cancer survivors as well.

## 1. Introduction

Gastric cancer is the fourth leading cause of cancer-related mortality worldwide, and its 5-year survival rate is less than 40%. *H. pylori* remains the leading cause of gastric cancer, which could vary from lymphomas, sarcomas, gastrointestinal stromal, and neuroendocrine tumors. With the development of biomarkers, more and more therapeutic strategies have been designed to treat gastric cancer. However, early detection of recurrent gastric cancer can fundamentally improve a patient’s survival, so it is essential to continue screening and monitoring even after the patient is disease-free. Early detection will allow physicians to provide early treatments for maximum benefit if cancer tumors recur in the stomach or elsewhere. Meanwhile, doctors and patients have always given importance to the issue of how to observe cancer recurrence with caution. In Taiwan, early detection and diagnosis have become feasible due to cancer screening promotion in recent years.

Artificial intelligence (AI), with the improvement of computing capacity, has been paid attention to in the medical field. Machine learning (ML), which is part of AI, is critical in helping diagnosis and prognosis. Zhang et al. [1] developed a multivariate logistic regression analysis on a nomogram with radiomic signature and clinical risk factors to predict early gastric cancer recurrence. Liu et al. [2] used the Support Vector Machine classifier on the gene expression profiling dataset to predict gastric cancer recurrence and identify correlated feature genes. Zhou et al. [3] benchmarked the algorithms of Random Forest (RF), GBM, Gradient Boosting, decision trees, and Logistics. They concluded that the first four factors affecting postoperative recurrence of gastric cancer were body mass index (BMI), operation time, weight, and age.

We developed a risk prediction model for survivors of recurrent gastric cancer based on the above trends. In our study, we propose Random Forest [4] to develop a classifier and benchmark for Multilayer Perceptron (MLP) [5], C4.5 [6], AdaBoost [7], and Bootstrap [8] and Aggregation (Bagging [9]) algorithms in metrics. Regarding data preprocessing, we leverage Synthetic Minority Oversampling Technology (SMOTE) [10] oversampling against imbalanced dataset issues. For strategic risk assessment, we use cost-sensitive learning as a trade-off tool. Lastly, SHapley Additive exPlanations (SHAPs) [11,12,13] are used for feature importance analysis from both global and local perspectives.

Finally, RF with SMOTE on this dataset can perform well, and SHAPs can show reasonable interpretation and feature importance. Furthermore, the top five risk factors for recurrent gastric cancer were identified as stage, number of regional lymph node involvement, *Helicobacter pylori*, BMI, and gender.

## 2. Materials and Methods

### 2.1. Data Preparation and Machine Learning Models

A hospital cohort of 2476 patients diagnosed with gastric cancer survivors was enrolled in the Taiwan Cancer Registry (TCR) database from July 2008 to August 2020. Of them, 432 recurrent gastric cancers were used compared to 2044 non-recurrent survivors. All patients underwent curative surgery for gastric cancer, following standardized procedures for tumor resection and lymph node dissection. Therefore, there was no subgroup of patients who underwent local excision surgery. Our database was analyzed in accordance with the *Taiwan Cancer Registry Coding Manual Long Form Revision* in 2018. This manual is published and revised by the Health Promotion Administration, Ministry of Health and Welfare. All hospitals in Taiwan register cancer data using this manual to provide information to the Taiwan Cancer Registry Center for integration. All clinical figures recorded in this database were evaluated and used to establish our predictive model. These clinical figures of gastric cancer survivors were used as the predictive features. In this study, the proposed process flow diagram is illustrated in Figure 1.

In step 1, we collected a recurrent gastric cancer dataset. For a better fit to our machine learning algorithms and TreeSHAP analysis, we encoded all our data into categorical variables in step 2. Due to the data imbalance between the non-recurrence and recurrence categories, we balanced the dataset with using upsampling method, SMOTE, for the minor category as step 3. One of our essential purposes is to understand the prediction behaviors of our trained model; therefore, we used 10-fold cross-validation without a dataset split in step 4. RF is the focus that is used to compare other baseline algorithms. In the final step, step 5, we further utilized model interpretation to observe feature importance and interactions.

RF is a kind of ensemble machine learning model based on Classification and Regression Trees (CART), Bagging, and random feature selection. This randomness design is suitable for preventing the overfitting of other decision trees and tolerance against noise and outliers. The tree elements make decisions based on information gained to obtain automatic feature selection. Namely, RF has a built-in feature selection function within the training phase, so there is no need to prefilter all features via principal component analysis. We can also interpret that selection in the feature importance analysis section. The importance of RF features is a permutation approach that measures the decrease in prediction performance as we permute the value of the feature.

RF has become one of the most popular machine learning algorithms based on the above properties. It has several advantages: (1) It is more accurate due to the ensemble approach. (2) It works without detailed hyperparameter setting and principal components analysis (PCA) preprocess. (3) It computes efficiently and quickly. These are the reasons we chose RF as our backbone algorithm.

SMOTE (Synthetic Minority Oversampling Technology) is an oversampling algorithm proposed by Chawla et al. to improve overfitting. The method creates new randomly generated samples between samples of minority classes and their neighbors, which can balance the number among categories.

Cost-sensitive learning [14,15,16] is a training approach that considers the assigned costs of misclassification errors while training the model. This is closely related to the imbalanced dataset. We need different biases when we need other monitoring criteria in various risk management phases. The computing speed of RF is efficient, so we can easily integrate different cost biases for interested scenarios to obtain an overall picture. In this study, we refer to a machine learning approach that considers the costs associated with different types of classification errors. In medical contexts, misclassifying specific outcomes can have significant consequences, both in terms of patient health and healthcare resources. Therefore, the goal of cost-sensitive learning in medical informatics is to optimize the classification model in a way that minimizes the overall cost, which may include considerations such as misclassification costs, treatment costs, patient preferences and risks, and medical resource allocation. By incorporating these considerations into the learning process, cost-sensitive learning in medical informatics aims to develop more accurate and practical models that better align with real-world constraints and objectives. This can ultimately lead to improved patient outcomes and more efficient use of healthcare resources.

SHAP analysis is an extension based on Shapley values, which is a game–theoretical method used to calculate the average of all marginal contributions in all coalitional combinations. Unlike the calculation of Shapley values, SHAPs address the addictive features as a linear model. For a given feature set, SHAP values are calculated for each feature and added to a base value to provide the final prediction.

Global interpretation in SHAPs can provide insights into the overall importance of different features in the model. By calculating SHAP values for each feature across a large dataset, we can determine which features contribute the most to predictions on average. This helps us to understand which risk factors are most influential across the entire population. The waterfall plot helps visualize the impact of each feature on the model’s output across the dataset. These plots allow us to see the direction and magnitude of each feature’s effect on predictions, providing a comprehensive overview of the model’s behavior.

Local interpretation in SHAPs can also provide explanations for individual predictions, allowing us to understand why a particular prediction was made for a specific patient. By calculating SHAP values for each feature for a single prediction, we can see how each feature contributed to that prediction. The plot helps visualize the contribution of each feature to the difference between the model output and the average output. It provides a detailed breakdown of how each feature affects the prediction for a given instance, making it easier to understand the reasoning behind individual predictions.

Using SHAPs for both global and local interpretation in medical risk factor prediction can help clinicians and researchers gain valuable insights into how the model works, which features are most important, and why specific predictions are made. This transparency and interpretability is crucial for building trust in predictive models and for making informed decisions in clinical settings.

TreeSHAP is proposed as a variant of SHAPs for tree-based machine learning models, such as decision trees, RF, and gradient-boosted trees.

The importance of a SHAP feature is defined as the average of absolute SHAP values per feature for all instances. This focuses on the variance of model output, which is different from the importance of a permutation feature based on performance error. From it, we can know how the magnitude of model output changes, like for the likelihood or regression values, as we manipulate feature values, and it has nothing to do with performance, like accuracy or loss.

### 2.2. Dataset Sources

In the database of the Taiwan Cancer Registry, 17 variables are recorded as clinical potential features of recurrence: (1) gender; (2) age at diagnosis; (3) grade/differentiation; (4) tumor size; (5) number of regional lymph node involvement; (6) stage; (7) surgical margin involvement of the primary site; (8) radiation therapy; (9) chemotherapy; (10) BMI; (11) smoking; (12) chewing betelnut; (13) alcohol drinking; (14) value of the CEA carcinoembryonic antigen; (15) CEA test of carcinoembryonic antigen; (16) *Helicobacter pylori*; (17) lymphatic or vascular. Therefore, the analysis aimed to identify the most critical risk features of these 17 predictors.

Our database was analyzed in accordance with the *Taiwan Cancer Registry Coding Manual Long Form Revision* in 2018. This manual is published and revised by the Health Promotion Administration, Ministry of Health and Welfare. All hospitals in Taiwan register cancer data using this manual to provide information to the Taiwan Cancer Registry Center for integration. Therefore, the research database is a part of the national cancer registration database. For example, we collected detailed information on smoking (code-36) and subcodes (00-99, 00-99, and 00-99) on current smoking status (the first subcode), smoking history (the second subcode), and time since quitting smoking (the third code) according to the manual. In the initial analysis, participants were categorized into smoking and non-smoking groups regardless of their current smoking status. However, since smoking was not found to be a significant factor in the analysis, detailed smoking data are not included in this article. The original data contains detailed information on smoking duration, smoking cessation, and the time of cessation according to the Taiwan Cancer Registry Coding Manual Long Form Revision in 2018. In the preliminary analysis, participants were initially grouped into smoking and non-smoking categories, regardless of their smoking cessation status. However, subsequent analysis indicated that smoking was not a significant factor. Consequently, the detailed smoking data were not included in the article. All patients underwent curative surgery for gastric cancer, following standardized procedures for tumor resection and lymph node dissection.

The encoding and sample size features are organized in Figure 2. The original dataset has significantly imbalanced issues; that is, the minor category is about 20% of the majority. As mentioned above, the RF and Decision Tree will automatically select the best feature for each decision split. Therefore, we did not utilize PCA for dimensionality reduction.

### 2.3. Data Preprocessing

To better fit our machine learning algorithms and TreeSHAP analysis, we encode all of our data into categorical variables with an assigned integer. We also assign ordinal variables that are positive or have higher intensity with larger integers so that TreeSHAP can display them effectively using a trend chart or dependence plot. However, most features have missing or unavailable values that have been encoded as middle or average rank to minimize possible bias on feature importance trends. That is, the moderate impact of the NA category should be between the maximum and minimum of ordinal values.

### 2.4. Dataset Balancing

Due to a significantly imbalanced dataset, the model would tend to overfit the major categories and ignore the learning features of minorities. Generally, several approaches could overcome this learning bias due to target loss function design, such as assigning different weights for samples or categories in the loss function, assigning additional cost weight for prediction errors, resampling data by over- or undersampling, etc.

SMOTE is the method we use for oversampling in this study. Instead of simply duplicating samples, we generate synthetic samples for minorities up to a quantity of the majority. It would select some nearby samples around a base sample of minority, randomly choose one neighbor, and randomly perturb one feature at a time within the distance between them.

### 2.5. 10-Fold Cross-Validation

In this study, the 10-fold cross-validation is used for performance checks. We use 10-fold cross-validation to prevent bias from a split of the whole dataset. Partition the complete dataset randomly into ten equal-sized subgroups. Each time, a subgroup will be chosen as a holdout set, and the rest of the night nine subgroups are for training. At the end of 10 training rounds, an average performance of 10 models will be output. We then perform the aggregation of various ordered feature importance lists based on the ranks with additional weights via the cross-entropy Monte Carlo algorithm using the Spearman distance. For each classifier, we perform upsampling to alleviate the class imbalance problem after inputting the training data. The classification metrics of the Random Forest algorithm, such as TP rate, FP rate, Precision, Recall, F1 score, and Accuracy, are illustrated in Table 1 below.

Then, we benchmarked RF using MLP, C4.5, AdaBoost, and Bagging for the machine learning algorithm.

MLP: A classifier that uses backpropagation to learn a Multilayer Perceptron to classify instances.C4.5: This algorithm develops a decision tree by splitting the value of the feature at each node, including categorical and numeric features. We calculated the information gain and used the feature with the highest gain as the splitting rule.AdaBoost with C4.5: It is a part of the group of ensemble methods called boosting and adds newly trained models in a series where subsequent models focus on fixing the prediction errors made by previous models. In this study, we selected C4.5 as the base classifier.Bagging (Bootstrap Aggregation) with C4.5: This is an ensemble skill that uses the bootstrap sampling technique to form different sets of samples with replacement. We used C4.5 as a base classifier to derive the forest.

The RF classification model develops parallel decision trees that vote on the category judgment for a given instance and output the final decision as a prediction. Cost-sensitive learning is essential in the case of risk management as we pursue better flexibility of trade-offs among metrics. For example, we may focus more on the recall rate of the recurrence category by loosening the performance of the precision rate. In this part, we assign different costs of the false negative error of recurrence categories 1, 2, 3, and 5 but keep the false positive error cost at 1.

In our study, we set the recurrence category as positive and then evaluated the metrics. True Positive (TP): number of positive instances predicted as positive. Negative (TN): number of negative instances predicted as negative. False Positive (FP): number of negative instances predicted as positive. False Negative (FN): number of positive instances predicted as negative. Accuracy: (TP + TN)/(TP + TN + FP + FN). Precision: TP/(TP + FP). Recall: TP/(TP + FN). F1-score: 2 × (Recall × Precision)/(Recall + Precision). False Positive Rate: FP/(FP + TN). True Positive Rate: TP/(TP + FN). The ROC curve (receiver operating characteristic curve) uses the True Positive Rate as the y-axis and False Positive Rate as the x-axis and plots points with corresponding thresholds.

### 2.6. Interpretability in Machine Learning Models

We review the importance of features using two approaches, RF and SHAP. The former is a single-feature permutation approach to observe model performance impact, whereas the latter is flexible in observing main features and the interaction effects regarding model output.

Concerning SHAP, we use global interpretability plots. The feature importance plot lists the importance of all features in ascending order and uses color bars with positive or negative correlation coefficients. The bee swarm plot for feature importance provides vibrant SHAP values and output impact direction information of individual points in rich colors that can help users obtain critical insights quickly. The dependence plot helpfully shows the correlations and interactions between the two features and the SHAP value trends.

The local interpretability plot, the waterfall plot, is designed to demonstrate explanations for individual instances.

## 3. Results

### 3.1. Traditional Predictor Algorithms

Before we established our prediction model, we analyzed these clinical features using a traditional statistic method, which shown in the table below. Finally, we tried to establish a cost-effective prediction model for recurrent gastric cancer.

The relevant risk factors and statistical chi-square test results (Table 2) are as follows: F1 Gender (OR = 1.30, 95%CI: 1.05–1.62, *p* = 0.019), F3 Grade/Differentiation (OR = 0.38, 95%CI: 0.19–0.78, *p* ≤ 0.001), F4 Tumor Size (OR = 0.66, 95%CI: 0.37–1.18, *p* < 0.001), F5 Number of regional lymph node involvement (OR = 0.75, 95%CI: 0.44–1.26, *p* < 0.001), F6 Cancer Stage (OR = 0.19, 95%CI: 0.41–0.88, *p* < 0.001), F7 Residual tumor on edge of primary site (OR = 2.76, 95%CI: 0.99–7.67, *p* < 0.001), F8 Radiation therapy (OR = 0.49, 95%CI: 0.33–0.72, *p* ≤ 0.001), F9 Chemotherapy (OR = 0.33, 95%CI: 0.26–0.41, *p* = 0.003), F10 BMI (OR = 1.23, 95%CI: 0.71–2.11, *p* = 0.009), F11 Smoking (OR = 1.79, 95%CI: 0.41–7.77, *p* = 0.008), F13 Alcohol drinking (OR = 2.25, 95%CI: 0.68–7.40, *p* = 0.032), F14 SSF1 Carcinoembryonic antigen CEA test value (OR = 1.98, 95%CI: 1.42–2.56, *p* < 0.001), F15 SSF2 Carcinoembryonic antigen CEA difference value (OR = 3.89, 95%CI: 2.69–5.61, *p* < 0.001), F16 SSF3 *Helicobacter pylori* (OR = 2.29, 95%CI: 1.69–3.10, *p* < 0.001), and F17 SSF5 Lymphatic or vascular (OR = 0.10, 95%CI: 0.03–0.270, *p* < 0.001); they are statistically significant variables. Traditional methods could provide a preliminary evaluation of these clinical features. However, we cannot rank these features at all. For clinicians, we expect them to know more about the ranking and interaction among these clinical features. As a result, AI models can provide us an alternative solution, as needed.

### 3.2. Prediction Performance

The balanced dataset has original major category 2044 instances and oversampled minor category 2044 instances. The relevant risk factors and statistical chi-square test were performed as a preliminary survey. In total, 17 clinical features—gender, age, grade/differentiation, tumor size, number of regional lymph node involvement, cancer stage, residual tumor on edge of primary site, radiation therapy, chemotherapy, BMI, smoking, alcohol drinking, Betel nut chewing, carcinoembryonic antigen CEA test value, carcinoembryonic antigen CEA difference value, *Helicobacter pylori*, and lymphatic or vascular invasion—are statistically significant variables. In this study, 10-fold cross-validation was used to prevent biases from splitting the entire dataset. It randomly divides the entire dataset into ten subgroups of the same size. Each time, a subgroup is selected as a holdout set, and the rest of the nine subgroups are trained. At the end of 10 training rounds, the average performance of 10 models will be produced. Then, we perform the aggregation of various ordered lists of important features based on the ranks with additional weights via the Monte Carlo cross-entropy algorithm using the Spearman distance. For each classifier, we performed upsampling to alleviate the problem of class imbalance after inputting the training data. Table 3 summarizes the comparison of different algorithms, including MLP, C4.5, AdaBoost with C4.5, Bagging with C4.5, and RF. RF has a metric outperformance of F1 of 88.2%, ROC area of 95.2%, PRC of 95.4%, and 87.9% for the recurrence category. First, MLP shows better metric performance than C4.5 as a benchmark between the numerical-base algorithm and the categorical-base decision tree algorithm. Second, from the evidence that RF significantly surpasses the C4.5 and Bagging sets, we believe that the main improvement is from the randomness design of bootstrap subsampling and the choice of feature for splitting node. Third, compared with AdaBoost, the independent trees of RF show better ensemble synergy than the boosting ensemble that arranges trees of the forest in series.

From the comparison of the ROC curves in Figure 3a, RF has the largest 0.952 area, which means that the trade-off between TPR and FPR through threshold setting is relatively better than others, while C4.5 is the worst. As a result, we chose Random Forest to establish our prediction model.

In Table 4, we can see that the recall rate increases (that is, 90.5% to 96.1%) if we increase the penalty cost on the FN (false negative) error of the recurrence category but keep the FP error cost as one precision rate (that is, 86% to 74.2%); the overall accuracy (that is, 87.9% to 81.4%) is compromised accordingly. In addition, the performance metrics of the Random Forest algorithm with different costs are compared and listed in the Table 4 below. These include the TP rate, FP rate, Precision, Recall, F1 score, ROC area, PRC area, MSE (Mean Square Error), and Accuracy. This policy scenario means that we hope that potential recurrence patients are labeled as much as possible, with acceptable results of FP (false positive) patient misclassification. In Figure 3b, it also shows no noticeable difference between different costs.

### 3.3. Interpretability

Before diving into the explanation of features, we need to understand that model interpretability is not always equal to causality. It is essential to address that SHAP values do not provide causality but instead provide insights into how the model behaves from data learning.

First, we used RF to observe global characteristics and it shows that F-6/F-5/F-2 are the top three impact features in Figure 4a. However, these feature importances did not show a positive/negative impact on recurrent gastric cancer. Therefore, the SHAP was applied and its value shows more information in Figure 4b. Firstly, it agrees that ***F-6*** and ***F-5*** have a critical and positive impact, which means that higher features bring bigger recurrence probability; meanwhile, ***F-2*** falls to sixth with a negative impact. Moreover, the ***F-16*** also moves up to the third feature with a negative impact. The red bars in Figure 4b have a positive correlation coefficient.

Next, we further investigated the SHAP distribution of all instances in Figure 5a; the bee swarm plot, F-6, and F-5 show higher feature value instances (i.e., red points contribute positive SHAP values), mostly with higher SHAP values. On the contrary, the feature values negatively correlate with the SHAP values (i.e., blue points contribute positive SHAP values).

In Figure 5b, the breakdown of the dependence plot shows the SHAP value in relation to the main effects and interaction effects if we want to investigate the interaction between features further. The on-diagonal values are the main effect, whereas the off-diagonal values are interactions. From the upper left corner of Figure 5b, we can see that interactions between F-5/F-6 would bring a negative SHAP value. In other words, F-5/F-6 would have positive main SHAP values, respectively, from the diagonal, but the off-diagonal interaction value would somehow decrease the main values.

Global interpretation shows correlations between features and prediction in all samples, which cannot clearly explain the prediction specific to a specific sample. In our study, we found competitive performance of this predicting model by using the top 10 clinical features.

For local interpretation in SHAPs, we can provide explanations for individual predictions in the recurrent case in Figure 6a and non-recurrent case in Figure 6b, allowing us to understand why a particular prediction was made for a specific patient. In Figure 6a, it locally indicates the breakdown of individual SHAP values of different clinical features of the recurrent case; from that, we can clearly see how those feature forces bring the prediction probability f(x) up to 1. In contrast, in Figure 6b, it shows reverse forces of different clinical features of the non-recurrent case bringing f(x) down to 0. Most importantly, in everyday work, we can analyze individual patients using RF and local SHAP analysis, as shown in Figure 6, to understand the impacts of all clinical features.

As a result, we established a cost-effective prediction model for recurrent gastric cancer using 8 of 17 clinical features and the model has competitive efficiency and performance. In real world data, especially for malignancies, there are numerous clinical features that need to be evaluated in relation to cancer recurrence. Using this method, it could be possible to use several leading clinical features to establish a reliable prediction model with the same performance instead of using all clinical features.

## 4. Discussion

More than 60% of gastric cancer survivors experienced recurrence after curative resection for gastric cancer, especially within two years after surgery. The risk factors for the recurrence patterns of different clinical or pathological factors were supposed to lead to recurrences of gastric cancer [17]. However, how to predict the recurrence of gastric cancer is an ongoing issue.

Lo et al. [18] found that the most critical risk factors for recurrence in early gastric cancer are lymph node status and the size of the mucosal tumor. Compared to advanced gastric cancer, the prognosis for patients with early gastric cancer is excellent. In our study, the result of F-4 (tumor size)/F-5 (number of regional lymph node involvement)/F-6 (cancer stage) with positive SHAP values is similar to their report.

In 2009, Tokunaga et al. [19] described that a 5-year survival rate after curative gastrectomy is better in overweight patients compared to non-overweight patients. Being overweight was revealed to be an independent prognostic factor in patients with early gastric cancer; however, the reason has not yet been determined. In our prediction model, F-10 (BMI) is the fourth leading feature with a negative impact on the recurrence of gastric cancer. The results were similar.

In 2020, Zheng et al. [20] identified that the pathological tumor (pT) stage and the pathological nodal (pN) stage were significantly associated with prognosis of stage I gastric cancer. The postoperative chemotherapy adjuvant might help improve the outcomes of high-risk patients. In our study, more clinical figures were collected and divided into subgroups for analysis. F-6 (cancer stage) is the leading risk factor, followed by F-5 (number of regional lymph node involvement). A similar result was concluded.

*Helicobacter pylori*, which is not only the leading risk factor for gastric cancer but also a high risk of recurrence, colonizes the gastric mucosa and induces persistent chronic gastric inflammation [21,22]. Patients with a genotype of high IgG1 will have a higher risk of recurrence than patients with other genotypes. Our study demonstrated that SSF3 (*Helicobacter pylori*) is the third leading feature for the recurrence of gastric cancer.

Artificial intelligence has become a workhorse for cancer diagnosis and prognosis with unprecedented accuracy, which is more powerful than traditional statistical analysis [23]. Chang et al. used a stacked ensemble-based classification model to predict the second primary cancer of head and neck cancer survivors by clinical features [24]. Using artificial intelligence, it is possible to develop a prediction model to determine clinical risk features, which will help clinicians screen cancer survivors before recurrence occurs.

In this study, we explore data mining using the machine learning algorithm RF on an imbalanced dataset. Furthermore, we utilize SMOTE to oversample the minority to balance and prevent the model from being biased too much on the original majority of non-recurrent instances. Regarding metric performance, RF shows better prediction capability than MLP, C4.5, AdaBoost, and Bagging. The prediction performance metrics can reach an overall accuracy of 87.9%, a recall rate of 90.5%, a precision rate of 86%, and an F1 of 88.2% in the recurrent category using 10-fold cross-validation in the balanced dataset.

Regarding cost-sensitive learning, as we increase the cost of FN, the recall rate can improve from 90.5% to 96.1%. Meanwhile, the precision rate is compromised from 86% to 74.2%, and the accuracy from 97.9% to 82.4%. Cost learning is a quick way of conventional machine learning to assess risk, while we intend to switch between different policies.

From SHAP value interpretation analysis, we can obtain insights into how models make a decision based on features and the interaction between correlated features. The identified top-five features are F-6 (cancer stage), F-5 (number of regional lymph node involvement), SSF3 (*Helicobacter pylori*), BMI, and gender. But, remember that model interpretation is for model behavior analysis and is not equal to causal effect analysis. The model learns from complicated correlations within the dataset, as shown in Figure 5b, and looks for the best choice to optimize an objective function.

In this study, we aimed to establish a cost-effective predicting model by using fewer clinical features. Although global interpretation shows correlations between features and prediction in all samples, which cannot clearly explain the prediction specific to a specific sample. However, in our study, we found, even by using eight clinical features, equal performance could be achieved by using our predicting model with SHAPs. For RF, we could figure out the importances of these clinical features. However, we could not realize the direction of them. Through SHAP values, the model is able to understand the positive or negative impact of these clinical features. Finally, we were able to identify the contributory risk factors of recurrent gastric cancer.

As a result, we established a cost-effective prediction model for recurrent gastric cancer using 17 clinical features and the model has competitive efficiency and performance. In real world data, especially for malignancies, there are numerous clinical features that need to be evaluated in relation to cancer recurrence. By using this method, we are able to use several leading clinical features to establish a reliable prediction model with the same performance instead of using all clinical features.

We have some limitations in this study. Firstly, data from the Taiwan Cancer Registry were collected from different hospitals and time periods. Some features have a significant number of missing values, such as lymphatic or vascular invasion. This situation may bias real trends if this feature contributes a significant SHAP value. We need to pay attention once we interpret this feature. Although there are 17 clinical features included, some risks such as surgical procedures, neoadjuvant regimens, races, and duration of smoking could not be assessed using this dataset and this could be a limitation.

## 5. Conclusions

In conclusion, RF is the best classifier of prediction capability in our study. We conclude RF with SMOTE on this dataset can reach outstanding performance and SHAPs can show good interpretation and feature importance; finally, we identify the top-five risk factors. Cost-sensitive learning could be achieved with an improvement in the recall rate from 90.5% to 96.1%, compromised precision rate from 86% to 74.2%, and accuracy from 97.9% to 82.4%. Mostly, F-6 (cancer stage), F-5 (number of regional lymph node involvement), SSF3 (*Helicobacter pylori*), BMI, and gender were the leading impact factors for recurrent gastric cancer. They will be helpful for physicians in detecting high-risk patients early in gastric cancer survivors.

This study used data from the Taiwan Cancer Registry Database to estimate retrospective clinical figures of gastric cancer at diagnosis and to evaluate the association with gastric cancer recurrence after the launch of targeted therapies. Despite these limitations, this study should provide an essential basis for further research.

## Figures and Tables

**Figure 1 diagnostics-14-00842-f001:**
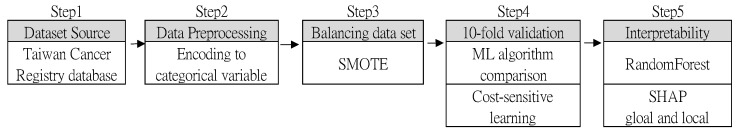
The proposed process flow diagram of this study.

**Figure 2 diagnostics-14-00842-f002:**
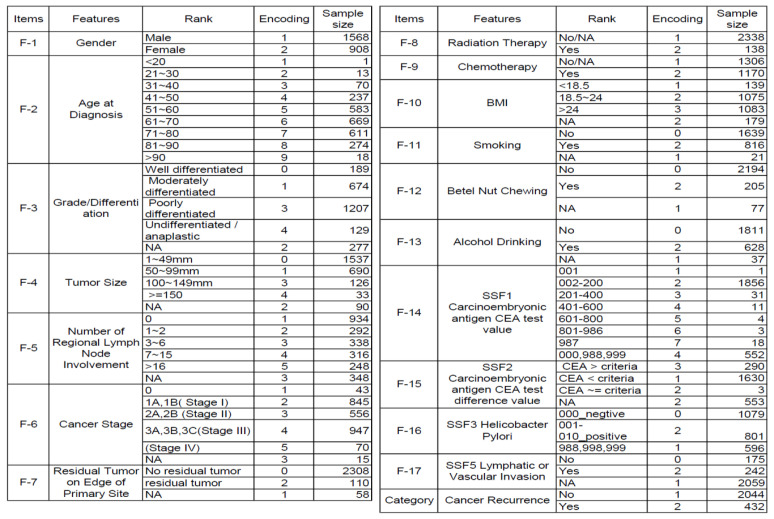
The dataset features with encodings and sample sizes were demonstrated in our predicting analysis.

**Figure 3 diagnostics-14-00842-f003:**
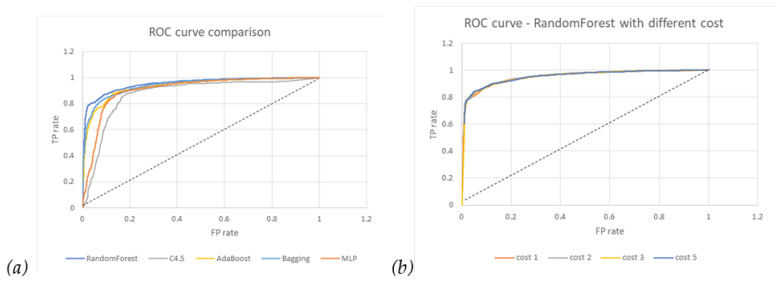
ROC curves: (**a**) different algorithms; (**b**) Random Forest with different FN costs.

**Figure 4 diagnostics-14-00842-f004:**
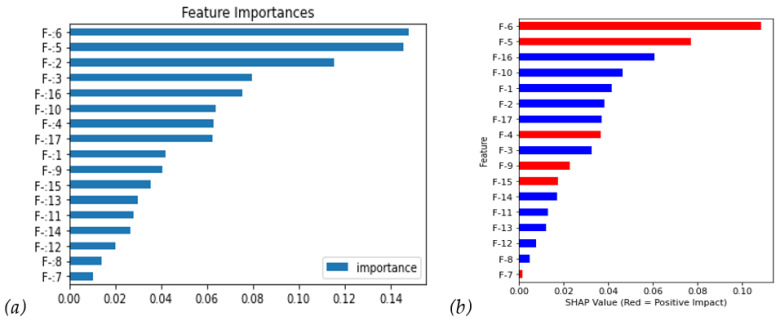
The clinical features were ranked by their feature importance: (**a**) Random Forest feature importance; (**b**) SHAP value with the costs of FN = 1. (Red: positive impact; Blue: negative impact).

**Figure 5 diagnostics-14-00842-f005:**
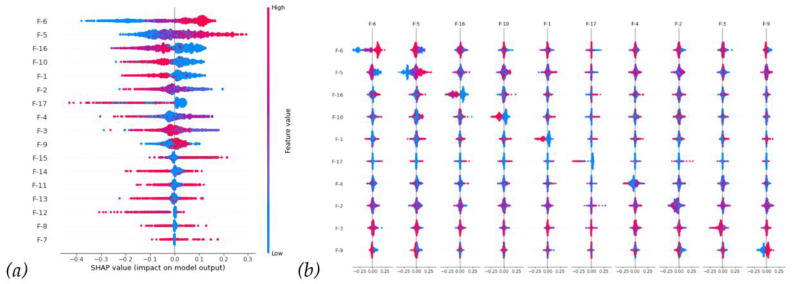
(**a**) Bee swarm plot. (**b**) Dependence plot. Included SHAP interaction matrix between the top 10 clinical features.

**Figure 6 diagnostics-14-00842-f006:**
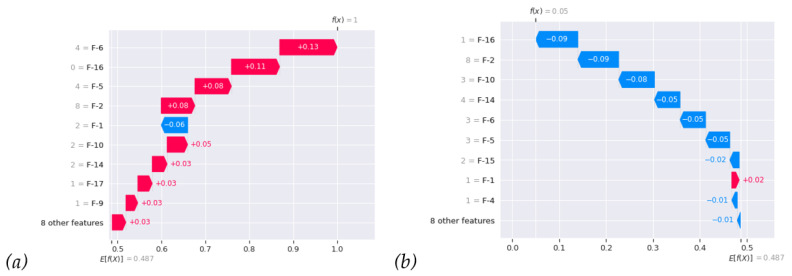
Waterfall plot. Two examples of local interpretation: (**a**) the recurrent case; (**b**) the non-recurrent case. f(x) is the expectation (that is, the average predictions of all instances).

**Table 1 diagnostics-14-00842-t001:** Classification Metrics of Random Forest derived from the confusion matrix.

TP Rate	FP Rate	Precision	Recall	F1 Score	Accuracy	Class
Average (Values of Random Forest)						
0.852	0.095	0.899	0.852	0.875	0.879	Non-Recurrence
0.905	0.148	0.860	0.905	0.882		Recurrence
Fold 1						
0.859	0.093	0.903	0.859	0.880	0.883	Non-Recurrence
0.907	0.141	0.864	0.907	0.885		Recurrence
Fold 2						
0.820	0.078	0.913	0.820	0.864	0.870	Non-Recurrence
0.922	0.180	0.836	0.922	0.876		Recurrence
Fold 3						
0.849	0.078	0.916	0.849	0.881	0.885	Non-Recurrence
0.922	0.151	0.858	0.922	0.889		Recurrence
Fold 4						
0.863	0.132	0.868	0.863	0.866	0.866	Non-Recurrence
0.868	0.137	0.863	0.868	0.866		Recurrence
Fold 5						
0.868	0.073	0.922	0.868	0.894	0.897	Non-Recurrence
0.927	0.132	0.876	0.927	0.900		Recurrence
Fold 6						
0.853	0.122	0.874	0.853	0.864	0.866	Non-Recurrence
0.878	0.147	0.857	0.878	0.867		Recurrence
Fold 7						
0.843	0.093	0.901	0.843	0.871	0.875	Non-Recurrence
0.907	0.157	0.853	0.907	0.879		Recurrence
Fold 8						
0.858	0.102	0.893	0.858	0.875	0.878	Non-Recurrence
0.898	0.142	0.864	0.898	0.880		Recurrence
Fold 9						
0.858	0.103	0.893	0.858	0.875	0.877	Non-Recurrence
0.897	0.142	0.863	0.897	0.880		Recurrence
Fold 10						
0.858	0.078	0.916	0.858	0.886	0.890	Non-Recurrence
0.922	0.142	0.866	0.922	0.893		Recurrence

**Table 2 diagnostics-14-00842-t002:** Traditional analysis on clinical features of recurrent gastric cancer.

		No Recurrence	Recurrence	Chi-Square Test	Hazard Ratio
		2044 (82.55%)	432 (17.44%)		
F1. Gender	Male	1273 (62.3%)	295 (68.3%)	5.541	1.00
	Female	771 (37.7%)	137 (31.7%)	(*p* = 0.019) *	1.30 [1.05–1.62]
F2. Age at Diagnosis	<20	1 (0.02%)	0 (0.0%)	6.389	1.00
	21~30	11 (0.53%)	2 (0.46%)	(*p* = 0.604)	0.00 [0.00- ]
	31~40	56 (2.73%)	14 (3.24%)		3.09 [0.24–38.31]
	41~50	186 (9.09%)	51 (11.81%)		4.25 [0.52–34.71]
	51~60	488 (23.9%)	95 (21.99%)		4.66 [0.60–35.86]
	61~70	558 (27.3%)	111 (25.69%)		3.30 [0.43–25.16]
	71~80	505 (24.7%)	106 (24.54%)		3.38 [0.44–25.67]
	81~90	222 (10.9%)	52 (12.04%)		3.56 [0.47–27.10]
	>90	17 (0.83%)	1 (0.23%)		3.98 [0.51–30.60]
F3. Grade/Differentiation	Well differentiated	178 (8.71%)	11 (2.55%)	40.698	1.00
	Moderately differentiated	557 (27.25%)	117 (27.08%)	(*p* ≤ 0.001) **	0.38 [0.19–0.78]
	Poorly differentiated	951 (46.53%)	256 (59.26%)		1.32 [0.88–1.96]
	Undifferentiated/anaplastic	119 (5.82%)	10 (2.31%)		1.69 [1.17–2.44]
	NA	239 (11.69%)	38 (8.80%)		0.52 [0.25–1.09]
F4. Tumor Size	1~49 mm	1357 (66.39%)	180 (41.67%)	103.840	1.00
	50~99 mm	492 (24.07%)	198 (45.83%)	(*p* ≤ 0.001) **	0.66 [0.37–1.18]
	100~149 mm	92 (4.50%)	34 (7.87%)		2.01 [1.12–3.58]
	>= 150	28 (1.37%)	5 (1.16%)		1.84 [0.93–3.64]
	NA	75 (3.67%)	15 (3.47%)		0.89 [0.29–2.68]
F5. Number of regional lymph node involvement	0	889 (43.49%)	45 (10.42%)	358.366	1.00
	1~2	235 (11.50%)	57 (13.19%)	(*p* ≤ 0.001) **	0.75 [0.44–1.26]
	3~6	254 (12.43%)	84 (19.44%)		3.59 [2.13–6.04]
	7~15	209 (10.23%)	107 (24.77%)		4.90 [2.98–8.05]
	>16	131 (6.41%)	117 (27.08%)		7.58 [4.64–12.39]
	NA	326 (15.95%)	22 (5.09%)		13.23 [8.03–21.78]
F6. Cancer Stage	0	43 (2.10%)	0 (0.0%)	298.851	1.00
	1A, 1B (Stage I)	821 (40.17%)	24 (5.56%)	(*p* ≤ 0.001) **	0.00 [0.00- ]
	2A, 2B (Stage II)	480 (23.48%)	76 (17.60%)		0.19 [0.41–0.88]
	3A, 3B, 3C (Stage III)	640 (31.31%)	307 (71.06%)		1.02 [0.22–4.65]
	(Stage IV)	47 (2.30%)	23 (5.32%)		3.11 [0.69–13.90]
	NA	13 (0.64%)	2 (0.46%)		3.18 [0.66–15.29]
F7. Residual tumor on edge of primary site	No residual tumor	1916 (93.73%)	392 (90.74%)	22.657	1.00
	residual tumor	74 (3.62%)	36 (8.33%)	(*p* ≤ 0.001) **	2.76 [0.99–7.67]
	NA	54 (2.64%)	4 (0.93%)		6.56 [2.20–19.55]
F8. Radiation therapy	No/NA	1946 (95.21%)	392 (90.74%)	13.508	1.00
	Yes	98 (4.79%)	40 (9.26%)	(*p* ≤ 0.001) **	0.49 [0.33–0.72]
F9. Chemotherapy	No/NA	1173 (57.39%)	133 (30.79%)	101.243	1.00
	Yes	871 (42.61%)	299 (69.21%)	(*p* ≤ 0.001) **	0.33 [0.26–0.41]
F10. BMI	<18.5	107 (5.23%)	32 (7.41%)	11.498	1.00
	18.5~24	869 (42.51%)	206 (47.69%)	(*p* = 0.009) *	1.23 [0.71–2.11]
	>24	924 (45.21)	159 (36.80%)		0.97 [0.65–1.45]
	NA	144 (7.05)	35 (8.10%)		0.70 [0.47–1.06]
F11. Smoking	No	1378 (67.42%)	261 (60.42%)	9.588	1.00
	Yes	647 (31.65%)	169 (39.12%)	(*p* = 0.008) *	1.79 [0.41–7.77]
	NA	19 (0.93%)	2 (0.46%)		2.48 [0.57–10.75]
F12. Betelnut Chewing	No	1803 (88.21%)	391 (90.51%)	2.467	1.00
	Yes	173 (8.46%)	32 (7.41%)	(*p* = 0.291)	1.63 [0.81–3.31]
	NA	68 (3.33%)	9 (2.08%)		1.39 [0.63–3.08]
F13. Alcohol drinking	No	1510 (73.87%)	301 (69.68%)	6.855	1.00
	Yes	500 (24.46%)	128 (29.63%)	(*p* = 0.032) *	2.25 [0.68–7.40]
	NA	34 (1.66%)	3 (0.69%)		2.90 [0.87–9.59]
F14. SSF1 Carcinoembryonic antigen CEA test Value	001	1 (52.9%)	0 (0.0%)	51.726	1.00
	002~200	1511 (73.92%)	345 (79.86%)	(*p* ≤ 0.001) **	0.00 [0.00- ]
	201~400	21 (1.03%)	10 (2.31%)		1.98 [1.42–2.56]
	401~600	5 (0.24%)	6 (1.39%)		3.97 [1.78–8.85]
	601~800	2 (0.15%)	2 (0.46%)		10.02 [2.96–33.86]
	801~986	1 (0.05%)	2 (0.46%)		8.35 [1.15–60.42]
	987	10 (0.49%)	8 (1.85%)		16.71 [1.49–187.1]
	000,988,999	493 (24.12%)	59 (13.66%)		6.68 [2.53–17.60]
F15. SSF2 Carcinoembryonic antigen CEA difference Value	CEA > criteria	198 (0.96%)	92 (21.30%)	59.362	1.00
	CEA < criteria	1349 (67.00%)	281 (65.05%)	(*p* ≤ 0.001) **	3.89 [2.69–5.61]
	CEA~ = criteria	3 (0.15%)	0 (0.0%)		1.74 [1.29–2.35]
	NA	494 (24.17%)	59 (13.66%)		0.00 [0.00- ]
F16. SSF3 *Helicobacter pylori*	000_negtive	852 (41.68%)	227 (52.55%)	30.285	1.00
	001–010_positive	658 (32.19%)	143 (33.10%)	(*p* ≤ 0.001) **	2.29 [1.69–3.10]
	988,998,999	534 (26.13%)	62 (14.35%)		1.87 [1.36–2.57]
F17. SSF5 Lymphatic or vascular	No	171 (8.36%)	4 (0.92%)	30.087	1.00
	Yes	198 (9.69%)	44 (10.19%)	(*p* ≤ 0.001) **	0.10 [0.03–0.270]
	NA	1675 (81.95%)	384 (88.89%)		0.96 [0.68–1.36]

* refers to statistically significant as *p* value < 0.05. ** refers to statistically highly significant as *p* value < 0.001. NA: Not applicable.

**Table 3 diagnostics-14-00842-t003:** Comparison of classification performance for different algorithms.

Algorithm	TP Rate	FP Rate	Precision	Recall	F1 Score	ROC Area	PRC Area	Accuracy	Category
MLP	0.835	0.112	0.882	0.835	0.858	0.909	0.91	0.862	Non-Recurrence
	0.888	0.165	0.843	0.888	0.865	0.909	0.883		Recurrence
C4.5	0.812	0.123	0.869	0.812	0.839	0.874	0.849	0.844	Non-Recurrence
	0.877	0.188	0.823	0.877	0.849	0.874	0.826		Recurrence
AdaBoost C4.5	0.859	0.115	0.882	0.859	0.87	0.933	0.924	0.872	Non-Recurrence
	0.885	0.141	0.863	0.885	0.873	0.933	0.937		Recurrence
Bagging C4.5	0.829	0.111	0.882	0.829	0.855	0.941	0.932	0.859	Non-Recurrence
	0.889	0.171	0.839	0.889	0.863	0.941	0.945		Recurrence
Random Forest	0.853	0.095	0.899	0.853	0.875	0.952	0.945	0.879	Non-Recurrence
	0.905	0.147	0.860	0.905	0.882	0.952	0.954		Recurrence

TP: True positive; FP: False positive.

**Table 4 diagnostics-14-00842-t004:** Comparison of the performance of the Random Forest classification performance for different costs.

Cost of FN	TP Rate	FP Rate	Precision	Recall	F1 Score	ROC Area	PRC Area	MSE	Accuracy	Category
1	0.853	0.095	0.899	0.853	0.875	0.952	0.945	0.176	0.879	Non-Recurrence
	0.905	0.147	0.860	0.905	0.882	0.952	0.954			Recurrence
2	0.799	0.066	0.924	0.799	0.857	0.954	0.948	0.186	0.866	Non-Recurrence
	0.934	0.201	0.823	0.934	0.875	0.954	0.955			Recurrence
3	0.743	0.058	0.928	0.743	0.825	0.953	0.947	0.199	0.842	Non-Recurrence
	0.942	0.257	0.785	0.942	0.857	0.953	0.953			Recurrence
5	0.666	0.039	0.945	0.666	0.782	0.953	0.947	0.221	0.814	Non-Recurrence
	0.961	0.334	0.742	0.961	0.838	0.953	0.954			Recurrence

TP: True positive; FP: False positive; FN: False negative.

## Data Availability

Data are available from the Institutional Review Board of Chung Shan Medical University Hospital for researchers who meet the criteria for access to confidential data. Requests for the data may be sent to the Chung Shan Medical University Hospital Institutional Review Board, Taichung City, Taiwan (e-mail: irb@csh.org.tw).

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
