# Peer review of "A Cost-Effective Model for Predicting Recurrent Gastric Cancer Using Clinical Features"

_diagnostics, 2024, doi:10.3390/diagnostics14080842_

Round 1

Reviewer 1 Report

Comments and Suggestions for Authors

The authors use a hospital-based cancer registry database that presents the incidence of recurrence and clinical risk features from 2476 gastric cancer survivors. The authors employed the Random Forest with MLP, C4.5, AdaBoost, and Bagging algorithms correcting results via SMOTE technique for imbalanced data, cost-sensitive learning for risk assessment, and SHAP method for feature importance analysis. The Random Forest appears outperform the other models resulting in ACC of 87.9%, a REC of 90.5%, and F1 of 88.2% using 10-fold cross-validation. They selected five clinical features of recurrent gastric cancer: stage, number of regional lymph node involvement, Helicobacter pylori, body mass index, and gender.

Comments:

      1)  Please explain the details of 10-fold cross-validation used (lines 213-214) where you determined mean values for quality metrics. Also, please provide variance values (MSE) of these metrics in such cross-validation.

     2) This reviewer did not understand the explication presented for fig.4 (b) Random Forest with different FN costs. Please provide the costs that are exposed in this graphic.

  3) This reviewer thinks that for better understanding by potential readers the ideas of this study, the authors should provide much more details in subsect. 3.2. Interpretability where some interpretation of SHAP values is provided. This reviewer proposes presenting the explication of the SHAP procedure. Please compare and explain the results presented in Figs. 4a, b where Random Forest feature importance and SHAP values are exposed. You wrote “Global interpretation shows correlations between features and prediction in all samples, which cannot clearly explain the prediction specific to a specific sample” (lines 265-266). How can the existing correlation affect the final selection of the features?

Reviewer 2 Report

Comments and Suggestions for Authors

Paper looking at using AI to develop an algorithm for prediction of recurrence of gastric cancer. Several Comments

1. The paper would be greatly strengthened by looking at comparison of the survival curves using the traditional staging methods with the AI prediction. As the AI only added BMI and HP status there appears to be little advantage over the traditional methods. It would also make it unlikely based on this that any further patients would be identified as high risk for closer surveillance based on this data. It would be useful if the actual numbers or percentage of extra patients whom were identified as high risk could be stated in this paper

2. my interpretation was that only adjuvant chemotherapy and radiotherapy was included. Was neoadjuvant treatment included in the data and was there any attempt to separate this from adjuvant. This would have specific implications jurisdictions where neoadjuvant  regimes are standard of care.

3. Obviously the paper is limited by the longitudinal data available within the registry but was there any way of further defining smoking according to pack years smoked or into cessation after surgery, or even when they stopped smoking. A registry which is simply smoking yes or no only records the status at the time of diagnosis and unfortunately many patients will say that they are non smokers when they stopped only a week before being seen.

4.  Similarly was there subgroup analysis looking at recurrence rates of those patients who had recurrence in residual stomach in those patients who had subtotal gastrectomies or local excisions as this could then inform surveillance intervals and recommendations

5. The authors need to acknowledge the limitation in applicability of this study in western cohorts. The vast majority of western cohorts have BMI greater than 24. The incidence of HP positivity is much smaller in western cohorts and the distribution of disease within the stomach varies significantly to asian cohorts. Similiarly the regimes of chemotherapy and radiotherpay differ between asian and western cohorts in timing and type and this may effect the use of AI.

6. Perhaps it may be more useful to suggest that AI may be used as a self learning tool  for each individual jurisdiction as a predictor according to the natures of the disease but that that this must be compared with the traditional predictor algorithms. 

Comments on the Quality of English Language

nil

Round 2

Reviewer 1 Report

Comments and Suggestions for Authors

The authors have attended all comments of this reviewer.

Reviewer 2 Report

Comments and Suggestions for Authors

the authors have met my concerns to my satisfaction  and is suitable for publication